# Anthocyanin Content and *Fusarium* Mycotoxins in Pigmented Wheat (*Triticum aestivum* L. spp. *aestivum*): An Open Field Evaluation

**DOI:** 10.3390/plants12040693

**Published:** 2023-02-04

**Authors:** Marco Gozzi, Massimo Blandino, Chiara Dall’Asta, Petr Martinek, Renato Bruni, Laura Righetti

**Affiliations:** 1Department of Food and Drug, University of Parma, Parco Area delle Scienze 27/a, 43100 Parma, Italy; 2Department of Agricultural Forest and Food Sciences, University of Turin, Largo Paolo Braccini 2, 10095 Grugliasco, Italy; 3Agrotest Fyto, Ltd., Havlíčkova 2787/121, 767 01 Kroměříž, Czech Republic

**Keywords:** secondary metabolism, biotic stress, pigmented wheat, deoxynivalenol, UHPLC-MS/MS

## Abstract

Twelve *Triticum aestivum* L. spp. *aestivum* varieties with pigmented grain, namely one red, six purple, three blue, and two black, were grown in open fields over two consecutive years and screened to investigate their risk to the accumulation of multiple *Fusarium*-related mycotoxins. Deoxynivalenol (DON) and its modified forms DON3Glc, 3Ac-DON, 15Ac-DON, and T-2, HT-2, ZEN, and Enniatin B were quantified by means of UHPLC-MS/MS, along with 14 different cyanidin, petunidin, delphinidin, pelargonidin, peonidin, and malvidin glycosides. A significant strong influence effect of the harvesting year (*p* = 0.0002) was noticed for DON content, which was more than doubled between harvesting years growing seasons (mean of 3746 µg kg^−1^ vs. 1463 µg kg^−1^). In addition, a striking influence of varieties with different grain colour on DON content (*p* < 0.0001) emerged in combination with the harvesting year (year×colour, *p* = 0.0091), with blue grains being more contaminated (mean of 5352 µg kg^−1^) and red grain being less contaminated (mean of 715 µg kg^−1^). The trend was maintained between the two harvesting years despite the highly variable absolute mycotoxin content. Varieties accumulating anthocyanins in the pericarp (purple coloration) had significantly lower DON content compared to those in which aleurone was involved (blue coloration).

## 1. Introduction

*Triticum* species went through a long domestication process and subsequent intensive breeding [1]. The resulting large intraspecific variability of cultivated wheat, along with the contribution of wild relatives, guarantees the availability of multiple traits of agronomic, dietary, and even gastronomic relevance [2]. Such biodiversity is constantly exploited to breed new cultivars needed to provide adequate yields, innovative technological or nutritional properties, and to improve wheat resistance to biotic and abiotic stress.

Within the available germplasm, pigmented *Triticum aestivum* L. spp. *aestivum* lines recently emerged for multiple reasons. These varieties are reputed as good sources of bioactive compounds, mainly carotenoids, phlobaphenes, phenolic acids, anthocyanins, and other polyphenols valuable for their contribution to human health [3]. In agricultural practice, varieties with red grain controlled by dominant *R-1* alleles located in the long arm of hexaploid wheat chromosomes 3A, 3B, and 3D (*R-A1*, *R-B1*, and *R-D1*, respectively) are widely used. This grain colour is determined by catechin and proanthocyanidin [4]. Recessive alleles cause white tinging of the grain. In addition to varieties with red and white grain, there are forms with the presence of anthocyanins. Belonging to this group are wheat genotypes with a purple pericarp, which is conditioned by *Pp* genes. According to Tereshchenko et al. [5], a dark purple pericarp results from complementary action of the dominant alleles, *Pp-D1* (7D) and *Pp3* (2A). Presence of the dominant alleles, *Pp-A1* and *Pp3*, results in light purple colour, and the grain will remain uncoloured in the presence of the recessive *Pp3* allele even if two others alleles (*Pp-A1* and *Pp-D1*) are in the dominant state [6]. The complementary action of *Pp3* and *Pp1* genes was explained by the interaction of their products resulting in a functional regulatory complex for anthocyanin biosynthesis [7]. In addition to the purple pericarp, there is a blue aleurone conditioned by *Ba* genes (*Ba1*, *Ba2,* and *Ba3*) located in the fourth group of homologous chromosomes (4B, 4A and 4D, respectively) [8,9,10]. Since the genes can be jointed to each other, a black wheat genotype can be created by combining genes for blue aleurone and purple pericarp. In caryopses of pigmented wheat, more than forty different anthocyanins are selectively located in the pericarp or in the outer aleurone layers, giving the grain characteristic colours (purple, blue, black, or red) according also to their quality and quantity [11,12].

Overall, the accumulation of anthocyanins is considered a recently evolved trait in wild relatives of wheat, resulting from environmental adaptation to various forms of stress [13]. Their presence in common wheat is instead the consequence of multiple and independent gene transfers occurring during breeding from, among others, diploid *T. boeoticum* Boiss., tetraploid *T. turgidum* L. subsp. *abyssinicum* Vavilov/now *T. aethiopicum* Jakubz. var. *arraseita* (Hochst. and Körn.) Philat./, and decaploid *Thinopyrum ponticum* (Podp.) [11]. In planta, these secondary metabolites are involved in multiple roles, serving as a protection from frost, excessive UV exposure, osmotic disbalance, and drought. Their contribution is mostly related to their antioxidant properties and to the regulation of ROS-induced signalling cascades [14]. More recently, a high anthocyanin content was also linked to a reduced susceptibility to pathogen attacks in various crops and in fruits, both in pre- and post-harvest conditions, in agreement with the great flexibility of flavonoids [15]. Regarding phlobaphenes, but not yet for anthocyanins, a correlation between antifungal defence and accumulation of these secondary metabolites was recently suggested in maize with regard to mycotoxin accumulation related to *Fusarium* infection [16]. A larger body of evidence is instead available for sorghum, in which anthocyanins are actively involved in phytoalexin response against invading fungi [17]. Furthermore, since the consumption of wheat rich in anthocyanins requires the use of wholegrain flour, maintaining it is after milling the pericarp and/or the aleurone layer, it is crucial to verify the possible sanitary risk associate to the accumulation of contaminants, such as mycotoxins, which are generally located in the external kernel layers [18]. Fusarium head blight (FHB) is a major fungal disease in wheat, determining significant losses in yield and crop quality. FHB is caused mainly by *Fusarium graminearum* and *F. culmorum*, and is able to produce a variety of mycotoxins that accumulate in kernels, leading to harmful consequences after food and feed consumption [19]. *Fusarium* mycotoxins include, among others, trichothecenes such as deoxynivalenol (DON), T-2, HT-2, and other co-occurring toxins, such as enniatins (ENNs) and zearalenone (ZEN). Cereal crops can also biotransform many of the above-mentioned compounds to modified forms [20]. For instance, DON is transformed by adding a glucose moiety, giving rise to deoxynivalenol glucoside (DON3Glc), the most common and abundant masked mycotoxin [21].

As reported by Atanasova-Penichon et al. [22] and Etzerodt et al. [23], polar secondary metabolites with antioxidant properties may counteract toxigenic *Fusaria* and mycotoxin accumulation in wheat, although anthocyanin were not evaluated in depth in this regard. In further metabolomics studies, anthocyanins were deemed as resistance-related metabolites in barley and in wheat cultivar Sumai-3 [24,25]. Landoni et al. [16] found that the accumulation of anthocyanins and phlobaphenes is related to a lower contamination by fumonisin B_1_ in maize kernels. Finally, Choo et al. [26] demonstrated that black barley is more resistant to DON accumulation than yellow barley.

As reported for other secondary metabolites, anthocyanin biosynthesis and accumulation are strongly influenced by environmental factors, including climate, water availability, temperature, and ultimately growing seasons. This highlights the need for evaluations considering multiple harvests in the same location and a wider range of pigments [27].

An exact assessment of the effect of anthocyanins in wheat grain on mycotoxin accumulation would be possible if we compared genotypes imprinting each other only with individual genes responsible for individual grain colours. Near-isogenic lines derived from common spring wheat seeded Novosibirskaya 67 will be available in the future [28], with the possibility to also develop lines with different contents of anthocyanins.

This study is therefore aimed at investigating the susceptibility of pigmented wheat genotypes to *Fusarium*-related mycotoxins. To obtain reliable results, the investigation was carried out in open fields over a period of two years by comparing the anthocyanin profile and the accumulation of multiple mycotoxins in twelve wheat varieties of different colour and genetic background.

## 2. Results and Discussion

The two growing seasons showed a similar meteorological trend throughout the wheat crop cycle (Table 1), with the rainfall concentrated after sowing (November) and between stem elongation and flowering (April and May). The precipitation during the ripening stage (June) was higher in 2020 than 2019 (113 vs. 40 mm), while during flowering (May), the rainfall was similar between the two years. In addition, during flowering, 2020 had higher temperature than 2019, as described by differences in GDD data (Table 1).

We opted for a precise quantification and profiling of a wide range of glycosides and aglycones, including delphinidin and malvidin, that were seldomly evaluated in previous investigations. As expected, significant differences emerged between blue, purple, red, and black varieties, both from a quantitative and qualitative standpoint (Table 2). The total anthocyanins content for each cultivar and breeding line returns in Figure 1 as the deviation from the overall mean value. The dataset underwent a full factorial ANOVA, returning both the harvest year (*p* < 0.0001) and the cultivar or breeding line (*p* < 0.0001) as significant factors, while their interaction was not significant (*p* = 0.0672). Purple cultivar Anthograin^TM^ provided the best results in both harvest years (11171 and 4451 µg kg^−1^ in 2020 and 2019, respectively), with Ceraso (6676 and 1458 µg kg^−1^), Rosso (9690 and 2895 µg kg^−1^), AF Zora (9488 and 2810 µg kg^−1^), and KM 98-18 (6345 and 2650 µg kg^−1^) providing above-average contents. Overall, blue and black-grained wheats proved to be more varied in chemical composition, with the black ones showing quantities above LOQ for all 14 anthocyanins tested, while red and purple ones had only 9–10 types.

Overall, the pattern obtained was in accordance with previous reports, as purple-grained wheats were richer in peonidin and had limited content in petunidin, while malvidin glucoside was absent. Its presence was consistent in both blue and black ones, which were the sole varieties containing delphinidin glycosides [16]. Blue-grained wheats instead had higher amounts of cyanidin, malvidin, and petunidin, while red varieties were reported in the literature as the less rich in total anthocyanins and devoid of pelargonidin derivatives [11].

It must be noticed, however, that anthocyanin distribution within a single colour group was not uniform, and few exceptions were registered. For instance, despite being listed as purple varieties, Anthograin^TM^ and Ceraso differed for being rich in cyanidin, delphinidin and malvidin derivatives, which were absent or very scarce in other purple-grained wheats. Ceraso, despite being registered as a purple cultivar, interestingly had a phytochemical profile almost superimposable to blue ones, therefore showing a unique pattern. A large phytochemical variability must be then expected even within chromatic groupings. Additionally, several genes are responsible for the genetic basis of the purple colour of the grain [6,29], but detailed evaluation is often lacking in varieties that are under continuous development. The purple-grained wheats used in our experiment have an evidently different genetic determination of the pericarp colour, where Anthograin^TM^ is visually darker and AF Jumiko is lighter, and this is also consistent with Table 2. The genetic composition of *Purple pericarp* (*Pp*) alleles in the purple genotypes used in this study is unknown to us.

In absolute terms, the anthocyanin content for all varieties was lower than the literature data for the available varieties and with consistent differences between harvest years, which may be related to distinct agronomic and distinct environmental conditions, including mycotoxin exposure. However, if we refer to the 2019 and 2020 growing seasons, the agronomic cycle was not so different and, for instance, flowering of wheat took place after the same number of days from sowing. In particular, the significantly lower test weight between years, besides contributing to a dilution effect for anthocyanin content due to endosperm/pericarp ratio favouring the former, might be a sign of a different FHB pressure between 2019 and 2020.

Beyond the effects of environmental conditions on the anthocyanin profile, the high accumulation of mycotoxins, likely due to a relevant fungal infection, potentially altered biosynthetic pathways for anthocyanins in both harvest years. It was often reported, in other crops than wheat, that anthocyanins may increase as a consequence of *Fusarium* infection, albeit different behaviour was reported for scarcely and heavily infected plants [30]. It remains unclear if variations in anthocyanin profile and content could be a consequence of an alteration in plant biochemistry induced directly by mycotoxin exposure during the active part of the infection, or the result of previous elicitation exerted by biological exposure to *Fusarium*, e.g., by simple contact of wheat and fungal hyphae.

The whole potential array of compounds associated with *Fusarium* infection in wheat was screened, namely DON and its major modified form DON3Glc, 3Ac-DON, and 15Ac-DON, together with other relevant *Fusarium* mycotoxins, such as T-2, HT-2, ZEN, and Enniatin B (Table 3). The overall mycotoxins occurrence is within 471–9603 µg kg^−1^ for DON, <LOQ-357 µg kg^−1^ for DON3Glc, <LOD-98 µg kg^−1^ for ZEN, 22–951 µg kg^−1^ for Enniatin B. T-2, and HT-2 in all samples were <LOD or <LOQ, except for a sample of the variety KM 72-18 (year 2019), in which HT-2 was quantified at 21 µg kg^−1^ (data not shown). Acetylated forms of DON were not detected in any samples. Biotransformation rate, expressed as DON3Glc/TDON (where TDON is the sum of DON and DON3Glc), was in the range of 0.025–0.066 with a mean value of 0.047. Regarding the co-occurrence, solely DON and Enniatin B were detected in all the varieties analyzed.

Based on statistical analysis, results show a strong highly significant influence effect of the harvesting year (*p* = 0.0002) on DON content, which is more than doubled for the harvesting year 2020 (mean of 3746 µg kg^−1^ vs. 1463 µg kg^−1^). In addition, a striking influence of genotype with different grain colour on DON content (*p* < 0.0001) emerged as well, also in combination with the harvesting year (year*colour, *p* = 0.0091) with the blue grain found as the more contaminated (mean of 5352 µg kg^−1^) and red grain found as the less contaminated (mean of 715 µg kg^−1^) (Figure 2). This trend is also maintained between the two harvesting years, in spite of the highly variable absolute mycotoxin content.

This phenomenon is of particular interest because, to the best of our knowledge, it was reported so far. Both the effects mentioned above were found as significant according to ANOVA analysis and, particularly, a post-hoc Tuckey test showed a significant difference between blue and the other grain colours. The content of DON3Glc follows the trend described above for DON, with a higher DON3Glc content detected for the harvesting year 2020 (mean of 170 µg kg^−1^) and for blue grain (mean of 211 µg kg^−1^), even in that case we saw an effect of the harvesting year (*p* = 0.0021) and colour (*p* = 0.0240) very similar to what was reported for DON. We also found a positive correlation between the DON and DON3Glc content, considering all varieties for both years, based on Pearson’s correlation coefficient (r = 0.927, *p* < 0.0001), consistently with the current literature [31]. About the DON3Glc/TDON ratio, previously reported as correlated to resistance to FHB in durum and common wheat, the highest value was found for red grain (0.059) and the lowest for blue grain (0.038), consistently with what was observed regarding the contamination by DON [32]. However, this difference was not found as significant. Regarding the presence of Enniatin B, an emerging mycotoxin, we found an effect of the harvesting year (*p* = 0.001), with a greater contamination for the year 2020, but an influence of the genotype (grain colour) only for the year 2019. We also found a significant positive correlation between DON and Enniatin B content only for 2020 and not for 2019; this is probably due to the different chemical structure, metabolic pathway, and main *Fusarium* species involved in biosynthesis [33]. Finally, we were able to quantify ZEN in 32 out of 72 samples, of which 30 are from the harvesting year 2020, in accordance with a greater overall mycotoxin contamination. The maximum concentration detected for ZEN was 98 µg kg^−1^ for the blue grain cultivar Skorpion, year 2020.

During both years a correlation was observed between the chromatic typology of wheat and the presence of DON and its derivatives (Figure 2). Based on Pearson’s test, DON was positively correlated with total petunidin (r = 0.5338; *p* < 0.0001) and malvidin content (r = 0.4025; *p* = 0.0005) in wheat genotype, while no significant correlation was found with other compounds.

While observing the overall contamination, the blue and black varieties were found to be more contaminated, while the red and purple ones showed an overall lower occurrence of mycotoxin. While it would be incorrect to argue of a greater or lesser resistance to *Fusarium*, it is nevertheless possible to note a possible histological pattern for such data. In fact, in black and blue-grained wheats, anthocyanins are accumulated in the inner aleurone layer, whereas in purple types, these secondary metabolites are concentrated directly in the pericarp that is the outer layer directly exposed to the first contact with *Fusarium*. This distinct location potentially led to a different dynamic in exposure to the pathogen, or to a possible greater protection during the initial stages of infection in spikelet.

Blue-grained wheats in this study are also characteristically less resistant, and for instance, the cultivar Skorpion is considered extremely susceptible to *Fusarium* infection [34]. This hypothesis is seemingly reinforced by Tukey’s post-hoc test on DON, suggesting that the only really different group is that of blue grains. In our case, the wheats with black grain (AF Zora, KM 98-18) are the result of a combination of the *Ba2* gene and genes for purple pericarp. Black wheat, on the other hand, does not significantly differentiate from purple. In this case, it seems that in the black grain, the purple pericarp may have a protective role, saving the deeper blue aleurone against *Fusarium* attack. However, this hypothesis needs to be verified with further experiments to be able to compare near-isogenic lines, which differ only for the genes responsible for the anthocyanin biosynthesis, especially considering that the number of replicates considered in the present study is limited and post-hoc test may suffer from distortion with low sample size experiments. Furthermore, if the hypothesis is valid, it makes sense to further investigate the *Fusarium* resistance mechanisms among the current purple-grained wheats grain. This would lead to knowledge to drive breeding in order to create varieties with black-grained wheat with an enhanced resistance to mycotoxins accumulation. This is a point of reflection, given that both blues and blacks have a similar genetic, but blacks also have anthocyanins in the pericarp and are not limited to the inner aleurone layer. The fact that the blue-grained varieties conditioned by the *Ba2* gene in this study were susceptible to mycotoxin accumulation is a problem that may prevent their use in practice. It would be very appropriate to carry out a special study evaluating the importance of various donors carrying all individual *Ba* genes in terms of resistance to FHB or other factors causing plant stress. At the same time, the sole conventional red cultivar tested (Aubusson), despite providing the lowest amount of anthocyanins, was the less affected by mycotoxin contamination. This observation suggests that other defensive compounds than anthocyanins are involved in determining a greater resistance to mycotoxin accumulation for this cultivar.

Since the genes for the red, purple, and blue colour of the grain lie in different chromosomes, they can be combined with each other in the breeding process. A detailed study should be carried out with common varieties with red grain and with different degrees of resistance to FHB in relation to important phenolic compounds in the grain. For an exact evaluation of the effect of anthocyanins in wheat grain on mycotoxin accumulation, it should be appropriate to compare almost isogenic lines, differing from each other in individual genes for grain colour. This approach would make it possible to eliminate the interfering effects of unknown genes in the genetic background. Some near-isogenic lines for purple grain colouration were already created [35], while near-isogenic lines for blue colouration are also being gradually created [10]. It will be important that these lines are derived from the same recipient variety. It is clear that the presence of coloured substances in plant tissues has an evolutionary significance [36] and it is likely that the anthocyanin colouration of the grain affects mycotoxins accumulation in wheat. Further studies will be needed to confirm this hypothesis.

## 3. Materials and Methods

### 3.1. Chemicals

Analytical standards of DON and its acetylated forms of 3Ac-DON and 15Ac-DON (100 mg L^−1^ in acetonitrile), DON3Glc (50 mg L^−1^ in acetonitrile), T-2 toxin (100 mg L^−1^ in acetonitrile), HT-2 toxin (100 mg L^−1^ in acetonitrile), ZEN (100 mg L^−1^ in acetonitrile), and ENN B (1 g L^−1^ in methanol), were purchased from Romer Labs (Getzersdorf, Austria). UHPLC-grade methanol, acetonitrile, acetic acid, and water were purchased from VWR Chemicals (Radnor, PA, USA). Ammonium acetate was purchased from Sigma-Aldrich (St. Louis, MO, USA). Analytical standards of cyanidin 3-O-glucoside chloride, cyanidin 3-O-rutinoside chloride, malvidin 3-O-glucoside chloride, peonidin 3-O-glucoside chloride, pelargonidin 3-O-glucoside chloride, delphinidin 3-O-glucoside chloride, and petunidin 3-O-glucoside chloride were purchased from Extrasynthese (Genay Cedex, France).

### 3.2. Samples

Samples from twelve varieties of pigmented common wheat (*T. aestivum* L. spp. *aestivum* L.), grown in a sandy loam soil at Cigliano (NW Italy, 45°18′ N, 8°01′ E; altitude 237 m) were collected over two harvesting years (2019 and 2020). These varieties are characterized by four different grain colours (one red, chosen as a conventional control, 6 purple, 3 blue, and 2 black varieties) as described in Table 4 and in Appendix A.

The daily temperatures and precipitation were measured at a meteorological station near the experimental area. According to the wheat development, the daily temperatures were reported as growing degree days (GDD), using a 0 °C base value (Table 1). The agronomic technique commonly adopted in the area was applied. Briefly, the previous crop was maize, and the field was ploughed each year, incorporating the debris into the soil. Planting was conducted in 12 cm-wide rows at a seeding rate of 450 seeds m^−2^ in November. A total of 160 kg N ha^–1^ was applied, split equally at wheat tillering, growth stage (GS) 23, and at the beginning of the stem elongation (GS31). No fungicide was applied to control fungal diseases; in particular, at flowering (GS61-65) no fungicide treatment was carried out to control FHB infection. The sowing and harvest dates, together with the dates of the main GS, for each growing season are described by Scarpino and Blandino [37]. Treatments were assigned to an experimental unit using a completely randomized block design with three replicates. The plots measured 7 × 1.5 m.

The grain yields were obtained by harvesting the whole plot using a Walter Wintersteiger cereal plot combine harvester. The harvested grains were mixed thoroughly, and 4 kg grain samples were taken from each plot, ground using a ZM 200 Ultra Centrifugal Mill (Retsch GmbH, Haan, Germany) and representative sub-samples of each wholegrain flour were used directly to analyze the anthocyanin and the mycotoxin content. A further subsample of grain (500 g) was taken from each plot to determine the test weight (TW) using a GAC^®^ 2000 Grain Analyzer (Dickens-John Auburn, IL, USA).

### 3.3. Sample Preparation for Mycotoxin Analysis

Samples were extracted according to Malachová et al. [38] with some modifications; 2 mL of acetonitrile/water (80:20, *v*/*v*) mixture acidified with 0.1% formic acid was added to 0.5 g of ground wheat. The samples were extracted for 90 min using a platform shaker (Ika Werke, Breisgau, Germany) at a speed of 200 strokes/min and subsequently centrifugated for 2 min at 3000 rpm (radius 17.8 cm) at room temperature. Next, 1000 µL of supernatant were transferred into vials and then injected into the UHPLC-MS/MS system.

Three biological replicates and three technical replicates for each variety and each year were analyzed (N = 3 × 3 × 12 × 2 = 216).

### 3.4. UHPLC-MS/MS Mycotoxin Analysis

UHPLC-MS/MS analysis was carried out on UHPLC Dionex Ultimate 3000 coupled to a triple quadrupole mass spectrometer TSQ Vantage (Thermo Fisher Scientific, Waltham, MA, USA) equipped with an electrospray source (ESI). The chromatographic separation was obtained using a Sunshell column (Chromanik Technologies, Osaka, Japan) 2.1 × 100 mm, 2.6 µm particle size, heated to 40 °C. 2 µL of sample extract injected into the UHPLC system, and the flow rate was set up to 0.35 mL min^−1^. Gradient elution was performed by using water (eluent A) and methanol (eluent B), both acidified with 0.2% *v*/*v* CH3COOH. Ammonium acetate was added to the eluent A at the final concentration of 5 mM. Initial conditions were set at 98% A and 2% B for 2 min, then eluent B was increased to 20%. After an isocratic step (6 min), eluent B was further increased to 90%, and this condition was maintained for 10 min until the return to the initial condition. The total run time was 26.5 min. Mass spectrometric analysis was performed both in positive and negative ionization mode in multiple reaction monitoring (MRM), spray voltage 3000 V, capillary temperature 270 °C, vaporizer temperature 200 °C, sheath gas pressure 50 units, and auxiliary gas pressure 5 units. The following quantifier transitions were measured: DON m/z 355 > 295 (CE 13 eV) and m/z 355 > 265 (CE 19 eV), DON3Glc m/z 517 > 457 (CE 17 eV) and m/z 517 > 427 (CE 24 eV), 3Ac-DON and 15AcDON m/z 397 > 307 (CE 18 eV) and m/z 397 > 59 (CE 20 eV), T-2 m/z 484 > 215 (CE 19 eV) and m/z 484 > 185 (CE 22 eV), HT-2 m/z 442 > 263 (CE 11 eV), ZEN m/z 317 > 175 (CE 26 eV) and m/z 317 > 131 (CE 32 eV), ENN B m/z 640 > 528 (CE 20 eV), m/z 640 > 314 (CE 31 eV), m/z 640 > 214 (CE 26 eV), m/z 640 > 196 (CE 29 eV), m/z 640 > 186 (CE 37 eV), and m/z 640 > 86 (CE 44 eV). Calibration curves were set up using external standards (range 1 µg kg^−1^–2500 µg kg^−1^) for target analyte quantification. Data acquisition was performed by Thermo Xcalibur 2.2 software (Thermo Fisher Scientific, Waltham, MA, USA).

### 3.5. Extraction of Anthocyanins

Samples were extracted according to Zaupa et al. [39] with some modifications. For instance, 1.5 mL of ethanol/water/formic acid (97:2:1, *v*/*v*/*v*) mixture was added to 150 mg of ground wheat. The samples were extracted for 15 min using a platform shaker (Ika Werke, Breisgau, Germany) at a speed of 200 strokes/min and subsequently centrifugated for 10 min at 10,000 rpm (radius 9.5 cm) at 4 °C. Next, 1 mL of clear supernatant was evaporated to dryness using a centrifugal vacuum concentrator (Labconco, Kansas City, MO, USA) and re-dissolved in a mixture of 100 µL of methanol/water (50:50, *v*/*v*) and then transferred into vials prior to being injected into the UHPLC-MS/MS system.

### 3.6. UHPLC-MS/MS Anthocyanins Analysis

UHPLC-MS/MS analysis was carried out on the UHPLC Dionex Ultimate 3000, coupled to a triple quadrupole mass spectrometer TSQ Vantage (Thermo Fisher Scientific, Waltham, MA, USA) equipped with an electrospray source (ESI). The chromatographic separation was obtained using a Sunshell column (Chromanik Technologies, Osaka, Japan) 2.1 × 100 mm, 2.6 µm particle size, heated to 40 °C. Additionally, 2 µL of sample extract was injected into the UHPLC system and the flow rate was set up to 0.35 mL min^−1^. Gradient elution was performed by using water (eluent A) and acetonitrile (eluent B), both acidified with 0.1% *v*/*v* HCOOH. Initial conditions were set at 99% A and 1% B for 10 min, then eluent B was increased to 80%. After an isocratic step (4 min), the gradient was returned to the initial condition. The total run time was 23.5 min. Mass spectrometric analysis was performed in positive ionization mode in multiple reaction monitoring (MRM), spray voltage 3500 V, capillary temperature 270 °C, vaporizer temperature 300 °C, sheath gas pressure 50 units, and auxiliary gas pressure 10 units.

The following quantifier transitions were measured: cyanidin 3-*O*-glucoside m/z 449 > 287 (CE 30 eV), cyanidin 3-*O*-rutinoside m/z 595 > 287 (CE 30 eV), cyanidin *O*-malonyl-hexoside m/z 535 > 287 (CE 30 eV), delphinidin 3-*O*-glucoside m/z 465 > 303 (CE 30 eV), delphinidin *O*-rutinoside m/z 611 > 303 (CE 30 eV), peonidin 3-*O*-glucoside m/z 463 > 301 (CE 30 eV), peonidin *O*-rutinoside m/z 609 > 301 (CE 30 eV), peonidin *O*-malonyl-hexoside m/z 549 > 301 (CE 30 eV), petunidin 3-*O*-glucoside m/z 479 > 317 (CE 30 eV), petunidin *O*-rutinoside m/z 625 > 317 (CE 30 eV), malvidin 3-*O*-glucoside m/z 493 > 331 (CE 30 eV), pelargonidin 3-*O*-glucoside m/z 433 > 271 (CE 30 eV), pelargonidin *O*-rutinoside m/z 579 > 271 (CE 30 eV), and pelargonidin *O*-malonyl-hexoside m/z 519 (CE 30 eV). Calibration curves were set up using external standards (range 5 µg kg^−1^–6500 µg kg^−1^) for target analyte quantification. The concentration of cyanidin *O*-malonyl-hexoside was expressed as cyanidin 3-*O*-glucoside equivalents. The concentration of delphinidin rutinoside was expressed as delphinidin 3-*O*-glucoside equivalents. The concentration of peonidin *O*-rutinoside and peonidin *O*-malonyl-hexoside were expressed as peonidin 3-*O*-glucoside equivalents. The concentration of petunidin *O*-rutinoside was expressed as petunidin 3-*O*-glucoside equivalents. The concentration of pelargonidin *O*-rutinoside and pelargonidin *O*-malonyl-hexoside was expressed as pelargonidin 3-*O*-glucoside equivalents. Data acquisition was performed by Thermo Xcalibur 2.2 software (Thermo Fisher Scientific, Waltham, MA, USA).

### 3.7. Statistical Analysis

The statistical analysis was performed using IBM SPSS v.25.0 (SPSS Italia, Bologna, Italy). Three independent technical replicates were considered for each biological replicate, and three biological replicates were considered for each variety and each year. Data were log normalized prior to statistical analysis. ANOVA followed by Tukey post-hoc test (α = 0.05) and Pearson’s correlation test (α = 0.01) was run using Statistica 13.5.0.17 (Tibco Software Inc., Palo Alto, CA, USA).

## 4. Conclusions

The collected data highlighted a remarkable susceptibility of blue-pigmented wheat varieties and breeding lines to mycotoxin contamination, compared to genotypes with different anthocyanin content and histological placement in the grain. To enhance the use of wholegrain flour from blue-grained wheat in bakery products with superior antioxidant capacity, it will be necessary to substantially improve their susceptibility to mycotoxin accumulation.

Multiple reports suggested that anthocyanins may have a positive effect on oxidative stress, and also resistance against *Fusarium* biotic stress. Scaling up these evaluations to field-grown plants proved to be a challenge and the first results did not provide a clear and unequivocal correlation between anthocyanin content and mycotoxin accumulation, although the black-grained varieties of wheat, genetically similar to the blue one, but with anthocyanins also in the pericarp layer, showed lower mycotoxin content. A leading result, however, was that varieties and breeding lines accumulating these pigments in the pericarp (purple colouration) had significantly lower DON content if compared to those in which aleurone is involved (blue colouration). This evidence is relevant in terms of further experimental design. As histological distribution may affect the interplay between pigmented wheat and mycotoxin presence, further investigations should not look just at the phytochemical profile of pigmented wheat, but also at their histological properties and at the composition of individual alleles responsible for anthocyanin biosynthesis. Investigations with techniques capable of unveiling the spatial distribution of mycotoxins and plant defense metabolites in pigmented wheat kernels, such as MALDI mass spectrometry imaging, seem to be warranted.

## Figures and Tables

**Figure 1 plants-12-00693-f001:**
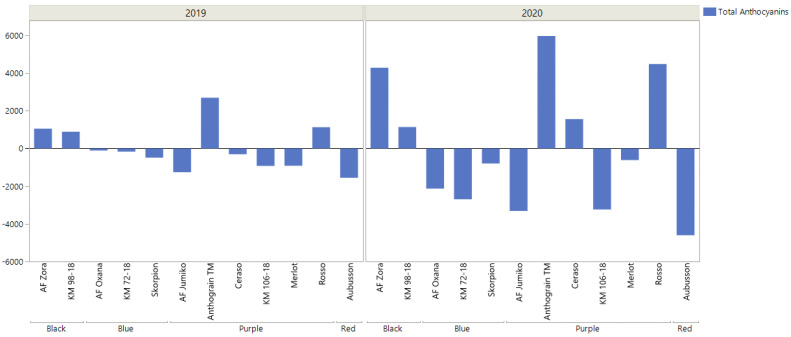
Deviation from the mean value calculated for total anthocyanins (expressed as the sum of all the metabolites detected within this study) for each variety and for each harvest year.

**Figure 2 plants-12-00693-f002:**
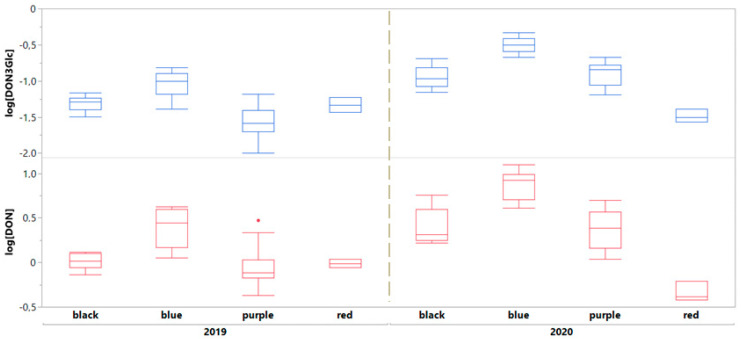
Box plot describing the DON and DON3Glc content in pigmented grains clustered according to the group and the harvest year. Data are log normalized.

**Table 1 plants-12-00693-t001:** Monthly rainfall, rainy days, and growing degree days (GDD) from the sowing (November) to the end of ripening stage (June) in the two growing seasons.

Growing Season	Month	Rainfall	Rainy Days	GDD ^1^
		(mm)	(days)	(Σ °C d^−1^)
2018–2019	November	124	5	292
	December	11	1	151
	January	6	1	141
	February	43	7	195
	March	17	4	314
	April	116	7	393
	May	178	9	478
	June	40	4	667
	November–June	535	39	2632
	November–March	200	19	1093
	April–June	335	21	1539
2019–2020	November	314	12	249
	December	132	8	193
	January	5	1	168
	February	1	0	229
	March	62	7	285
	April	81	10	414
	May	122	7	579
	June	113	7	624
	November–June	830	52	2741
	November–March	514	28	1124
	April–June	317	24	1617

Source: Rete Agrometeorologica del Piemonte—Regione Piemonte—Assessorato Agricoltura—Settore Fitosanitario, sezione di Agrometeorologia. ^1^ Accumulated growing degree days for each experiment using a 0 °C base value.

**Table 2 plants-12-00693-t002:** Occurrence of anthocyanins in twelve varieties of coloured common wheat. Three biological replicates and three replicates were considered for each variety. Data are reported as means ± SD. Different letters indicate significant differences (*p* < 0.005) among varieties within the same harvest year.

Year	Grain Colour	Variety	Cyanidin			Dephinidin		Peonidin			Petunidin		Malvidin	Pelargonidin			Total ACNs
			µg kg^−1^														
			3-O-Glc	3-O-Rut	O-MalHex	3-O-Glc	3-O-Rut	3-O-Glc	3-O-Rut	O-MalHex	3-O-Glc	3-O-Rut	3-O-Glc	3-O-Glc	3-O-Rut	O-MalHex	
2020	Black	AF Zora	735 ± 36 b	1867 ± 89 a	398 ± 92 c	622 ± 53 a	1041 ± 97 a	644 ± 149 b	1583 ± 345 a	1660 ± 409 b	127 ± 6 a	353 ± 27 a	122 ± 0	86 ± 11 b	99 ± 14 b	34 ± 4 c	9488 ± 930
		KM 98-18	415 ± 84 c	1983 ± 399 ab	225 ± 26 c	419 ± 51 b	1185 ± 18 9 a	241 ± 35 c	576 ± 78 c	597 ± 90 d	106 ± 5 a	387 ± 51 a	118 ± 2	27 ± 2 c	30 ± 2 de	<LOD	6345 ± 1012
	Blue	Skorpion	278 ± 48 d	1449 ± 240 b	93 ± 21 d	323 ± 28 c	838 ± 93 b	129 ± 20 d	420 ± 66 c	251 ± 55 e	99 ± 4 ab	293 ± 27 b	113 ± 1	32 ± 4 c	41 ± 7 d	<LOQ	4372 ± 630
		AF Oxana	146 ± 29 d	967 ± 213 c	29 ± 3 e	289 ± 37 c	817 ± 152 b	58 ± 4 d	236 ± 48 d	61 ± 6 f	90 ± 4 b	253 ± 42 b	111 ± 4	<LOQ	25 ± 1 e	25 ± 0 c	3074 ± 551
		KM 72-18	164 ± 45 d	752 ± 273 c	32 ± 8 e	255 ± 40 c	508 ± 123 c	66 ± 11 d	217 ± 50 d	74 ± 30 f	90 ± 5 b	197 ± 45 bc	109 ± 3	<LOD	27 ± 0 e	150 ± 27 b	2453 ± 591
	Purple	Anthograin^TM^ CDC	1362 ± 180 a	1459 ± 154 b	2194 ± 239 a	<LOD	<LOD	1354 ± 185 a	729 ± 81 b	3517 ± 341 a	<LOD	72 ± 0.2 c	111 ± 3	97 ± 9 b	64 ± 5 c	37 ± 2 c	11110 ± 1158
		Ceraso	451 ± 31 c	2077 ± 282 a	281 ± 19 c	412 ± 33 b	1211 ± 172 a	287 ± 24 c	593 ± 83 c	758 ± 90 c	104 ± 5 a	379 ± 51 a	118 ± 3	29 ± 1 c	31 ± 2 d	35 ± 2 c	6766 ± 720
		Jumiko	87 ± 9 e	124 ± 6 e	142 ± 16 d	<LOD	118 ± 5 d	273 ± 27 c	262 ± 19 d	735 ± 77 c	<LOD	74 ± 1 c	<LOD	26 ± 0 c	25 ± 0 d	50 ± 4 c	1891 ± 135
		KM 106-18	94 ± 14 e	206 ± 28 e	169 ± 12 d	<LOD	122 ± 6 d	224 ± 24 c	321 ± 41 d	650 ± 54 cd	<LOD	75 ± 2 c	<LOD	30 ± 2 c	30 ± 3 d	174 ± 42 b	1970 ± 185
		Merlot	244 ± 41 d	464 ± 74 d	277 ± 50 c	<LOD	111 ± 0 d	587 ± 122 b	842 ± 157 b	1665 ± 368 b	<LOD	75 ± 1 c	<LOD	77 ± 14 b	79 ± 14 bc	365 ± 16 a	4471 ± 823
		Rosso	571 ± 14 c	1443 ± 94 b	794 ± 34 b	<LOD	<LOD	1071 ± 20 a	1699 ± 82 a	3367 ± 104 a	<LOD	77 ± 1 c	<LOD	135 ± 3 a	167 ± 10 a	49 ± 9 c	9664 ± 348
	Red	Aubusson	28 ± 5 f	59 ± 32 f	25 ± 1 e	118 ± 0 d	125 ± 22 d	52 ± 1 d	57 ± 3 e	66 ± 4 f	<LOQ	80 ± 1 c	<LOD	<LOQ	<LOQ	212 ± 14 ab	437 ± 206
	*Mean ± SD*		*381 ± 376*	*1071 ± 742*	*388 ± 680*	*348 ± 158*	*608 ± 465*	*416 ± 422*	*628 ± 526*	*1117 ± 1215*	*103 ± 14*	*193 ± 133*	*115 ± 5*	*60 ± 40*	*56 ± 44*	*113 ± 12*	*5170 ± 3495*
			3-O-Glc	3-O-Rut	O-MalHex	3-O-Glc	3-O-Rut	3-O-Glc	3-O-Rut	O-MalHex	3-O-Glc	3-O-Rut	3-O-Glc	3-O-Glc	3-O-Rut	O-MalHex	Total ACNs
2019	Black	AF Zora	151 ± 14 b	413 ± 54 a	97 ± 25 c	214 ± 28 a	324 ± 67 ab	209 ± 51 b	509 ± 76 a	509 ± 145 b	78 ± 5	163 ± 35	95 ± 3	13 ± 4	13 ± 2	4 ± 2	2818 ± 161
		KM 98-18	129 ± 7 b	596 ± 52 a	111 ± 34 b	194 ± 18 ab	474 ± 43 a	125 ± 26 c	296 ± 14 bc	321 ± 97 c	75 ± 3	228 ± 14	96 ± 0	<LOQ	<LOQ	<LOQ	2650 ± 86
	Blue	Skorpion	38 ± 6 d	275 ± 57 b	7 ± 2 e	136 ± 5 b	252 ± 34 b	44 ± 4 d	166 ± 38 c	57 ± 14 e	70 ± 1	137 ± 19	92 ± 3	<LOQ	<LOQ	<LOQ	1275 ± 149
		AF Oxana	47 ± 3 d	404 ± 26 a	3 ± 1 e	175 ± 9 b	423 ± 20 a	37 ± 1 d	175 ± 8 c	38 ± 6 e	72 ± 1	185 ± 9	92 ± 1	<LOQ	<LOQ	<LOD	1639 ± 51
		KM 72-18	73 ± 36 c	393 ± 156 ab	3 ± 2 e	186 ± 45 ab	324 ± 96 ab	42 ± 5 d	187 ± 55 c	36 ± 3 e	78 ± 8	176 ± 49	96 ± 6	<LOQ	<LOQ	31 ± 8	1593 ± 461
	Purple	Anthograin^TM^ CDC	436 ± 56 a	381 ± 73 b	711 ± 64 a	<LOD	<LOQ	655 ± 70 a	329 ± 58 b	1789 ± 140 a	63 ± 2	<LOD	<LOD	22 ± 3	6 ± 2	65 ± 9	4451 ± 457
		Ceraso	65 ± 13 c	131 ± 21 c	118 ± 24 b	<LOD	88 ± 2 c	195 ± 34 bc	196 ± 39 c	572 ± 84 b	<LOQ	<LOD	<LOD	19 ± 3	11 ± 3	<LOQ	1458 ± 229
		Jumiko	14 ± 6 e	53 ± 3 d	28 ± 13 d	<LOD	89 ± 2 c	79 ± 25 d	56 ± 11 e	186 ± 77 d	<LOQ	<LOD	<LOD	<LOQ	<LOD	9 ± 1	504 ± 133
		KM 106-18	33 ± 5 d	73 ± 4 d	72 ± 9 c	<LOD	86 ± 0 c	123 ± 9 c	97 ± 9 d	345 ± 30 c	<LOQ	<LOD	<LOD	<LOQ	<LOQ	18 ± 2	780 ± 72
		Merlot	33 ± 3 d	77 ± 4 d	37 ± 3 d	<LOD	88 ± 2 c	147 ± 5 c	107 ± 7 d	333 ± 17 c	<LOQ	<LOD	<LOD	5 ± 0.4	<LOQ	83 ± 45	844 ± 41
		Rosso	167 ± 85 b	258 ± 122 bc	222 ± 109 b	<LOD	86 ± 0 c	381 ± 164 b	419 ± 206 ab	1238 ± 565ab	<LOQ	<LOD	<LOD	23 ± 13	17 ± 12	<LOQ	2866 ± 1276
	Red	Aubusson	<LOQ	49 ± 0 d	<LOQ	<LOD	86 ± 0 c	<LOQ	35 ± 0 e	39 ± 2 e	<LOQ	<LOD	<LOD	<LOQ	<LOD	59 ± 5	82 ± 112
	*Mean ± SD*		*108 ± 120*	*258 ± 182*	*128 ± 204*	*181 ± 29*	*211 ± 153*	*185 ± 185*	*215 ± 146*	*455 ± 540*	*73 ± 6*	*178 ± 33*	*94 ± 2*	*16 ± 8*	*12 ± 51*	*39 ± 31*	*1747 ± 1240*

Legend: 3-O-Glc: 3-*O*-glucoside; 3-*O*-Rut: 3-*O*-rutinoside; *O*-MalHex: *O*-malonyl-hexoside; ACNs: total anthocyanines.

**Table 3 plants-12-00693-t003:** Occurrence of mycotoxins in twelve varieties of coloured common wheat. Three biological replicates and three technical replicates were considered for each variety. Data are reported as means ± SD. Different letters indicate significant differences (*p* < 0.005) among varieties within the same harvest year.

Year	Grain Colour	Variety	DON	DON3Glc	3/15Ac-DON	ZEN	Enniatin B	T-2	HT-2
			mg kg^−1^						
2020	Black	AF Zora	2.327 ± 1.021 d	0.101 ± 0.034 d	<LOD	0.012 ± 0.000 d	0.247 ± 0.138 d	<LOQ	<LOD
		KM 98-18	3.280 ± 2.093 c	0.141 ± 0.060 c	<LOD	0.045 ± 0.014 b	0.518 ± 0.388 b	<LOQ	<LOQ
	Blue	Skorpion	7.235 ± 2.594 b	0.289 ± 0.060 b	<LOD	0.098 ± 0.047 a	0.418 ± 0.317 c	<LOQ	<LOQ
		AF Oxana	9.603 ± 2.767 a	0.357 ± 0.092 a	<LOD	0.088 ± 0.037 a	0.681 ± 0.314 b	<LOQ	<LOQ
		KM 72-18	6.769 ± 3.163 b	0.329 ± 0.103 a	<LOD	0.057 ± 0.020 b	0.951 ± 0.366 a	<LOQ	<LOQ
	Purple	Anthograin^TM^ CDC	1.466 ± 0.561 d	0.084 ± 0.010 d	<LOD	0.028 ± 0.013 c	0.281 ± 0.227 d	<LOD	<LOD
		Ceraso	2.333 ± 1.406 d	0.100 ± 0.054 d	<LOD	0.034 ± 0.007 c	0.554 ± 0.509 b	<LOD	<LOQ
		Jumiko	2.562 ± 0.271 d	0.177 ± 0.017 c	<LOD	0.023 ± 0.021 c	0.317 ± 0.192 c	<LOD	<LOD
		KM 106-18	3.889 ± 1.008 c	0.174 ± 0.033 c	<LOD	0.040 ± 0.032 b	0.416 ± 0.103 c	<LOQ	<LOQ
		Merlot	3.460 ± 0.494 c	0.152 ± 0.030 c	<LOD	0.032 ± 0.015 c	0.194 ± 0.071 de	<LOD	<LOD
		Rosso	1.555 ± 0.247 d	0.107 ± 0.028 d	<LOD	0.027 ± 0.014 c	0.129 ± 0.066 e	<LOD	<LOD
	Red	Aubusson	0.471 ± 0.128 e	0.033 ± 0.007 e	<LOD	0.01 ± 0.000 d	0.125 ± 0.057 e	<LOD	<LOD
	*Mean ± SD*		*3.746 ± 2.991*	*0.170 ± 0.109*	*n.d.*	*0.046 ± 0.033*	*0.403 ± 0.326*	*n.d.*	*n.d*
2019	Black	AF Zora	1.058 ± 0.176 cd	0.054 ± 0.013 b	<LOD	<LOD	0.083 ± 0.011 c	<LOD	<LOD
		KM 98-18	1.046 ± 0.299 cd	0.046 ± 0.013 c	<LOD	<LOQ	0.121 ± 0.068 b	<LOD	<LOQ
	Blue	Skorpion	3.212 ± 0.772 a	0.127 ± 0.026 a	<LOD	0.081 ± 0.000	0.130 ± 0.047 ab	<LOD	<LOD
		AF Oxana	3.922 ± 0.266 a	0.105 ± 0.018 a	<LOD	<LOD	0.103 ± 0.012 b	<LOD	<LOQ
		KM 72-18	1.377 ± 0.352 c	0.059 ± 0.021 b	<LOD	<LOQ	0.171 ± 0.043 a	<LOD	0.021 ± 0.000
	Purple	Anthograin^TM^ CDC	0.539 ± 0.121 f	0.030 ± 0.000 d	<LOD	<LOD	0.034 ± 0.012 d	<LOQ	<LOQ
		Ceraso	0.653 ± 0.199 e	<LOQ	<LOD	0.010 ± 0.000	0.026 ± 0.000 e	<LOD	<LOD
		Jumiko	0.752 ± 0.060 e	0.029 ± 0.006 d	<LOD	<LOD	0.044 ± 0.014 d	<LOD	<LOD
		KM 106-18	0.960 ± 0.361 d	0.033 ± 0.008 d	<LOD	<LOD	0.039 ± 0.005 d	<LOD	<LOD
		Merlot	2.039 ± 0.986 bc	0.053 ± 0.015 b	<LOD	<LOD	0.022 ± 0.000 e	<LOD	<LOD
		Rosso	1.040 ± 0.449	0.057 ± 0.013 b	<LOD	<LOD	0.038 ± 0.027 d	<LOD	<LOD
	Red	Aubusson	0.961 ± 0.114 d	0.054 ± 0.015 b	<LOD	<LOQ	0.058 ± 0.037	<LOQ	<LOQ
	*Mean ± SD*		*1.463 ± 1.096*	*0.053 ± 0.034*	*n.d.*	*0.052 ± 0.021*	*0.079 ± 0.011*	*n.d.*	*n.d.*

**Table 4 plants-12-00693-t004:** Pigmented wheat varieties compared in the two growing seasons field experiment.

Grain Color	Variety	Status	Registration	Responsible of Selection
Red (control)	Aubusson	Cultivar	2003	Limagrain Italia S.p.A., Fidenza (PR), Italy
Purple	Anthograin^TM^ CDC	Cultivar	2018	Hetland Seeds Ltd., Naicam, SK, Canada
	Ceraso	Cultivar	2014	Saatzucht Donau GesmbH. and CoKG, Austria
	AF Jumiko	Cultivar	2018	Agrotest Fyto, Ltd., Kroměříž, Czech Republic
	KM 106-18	Breeding line		Agrotest Fyto, Ltd., Kroměříž, Czech Republic
	Merlot	Cultivar	2015	Saatzucht Donau GesmbH. and CoKG, Austria
	Rosso	Cultivar	2011	Saatzucht Donau GesmbH. and CoKG, Austria
Blue	Skorpion	Cultivar	2011 ^1^	Agrotest Fyto, Ltd., Kroměříž, Czech Republic
	AF Oxana	Cultivar	2019	Agrotest Fyto, Ltd., Kroměříž, Czech Republic
	KM 72-18	Breeding line		Agrotest Fyto, Ltd., Kroměříž, Czech Republic
Black	AF Zora	Cultivar	2021	Agrotest Fyto, Ltd., Kroměříž, Czech Republic
	KM 98-18	Breeding line		Agrotest Fyto, Ltd., Kroměříž, Czech Republic

^1^ registered in Austria, since 2012 it has been included in the European list of varieties.

## Data Availability

Not applicable.

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
