# Peer review of "Anthocyanin Content and Fusarium Mycotoxins in Pigmented Wheat (Triticum aestivum L. spp. aestivum): An Open Field Evaluation"

_plants, 2023, doi:10.3390/plants12040693_

Round 1
Reviewer 1 Report
The manuscript contains only two figures for the results, one for the anthocyanin contents, shown as the deviation from the mean values, and another describing the DON and DON-3Glc contents. Beyond these, there are two tables, in a file named “non-published”, regarding the detailed anthocyanin and mycotoxin contents. This is far too low for a publication in a journal with IF 4.658. This is the most adverse issue, however, there are some minor ones too:
- showing of appearance of ears in a photo is a good idea but showing the different kernel colours would be better
- the statement of “a strong influence of the harvesting year” based on only two harvesting years does not sound serious
- in the second paragraph of the Introduction the terms “genes” and “alleles” are not used properly
- there is no data and explanation about the connection between anthocyanin contents and FHB (or any other sides of plant/pathogen interactions) in the Introduction
- in the “4.6 Statistical analysis” chapter is not clear that how many biological and technical replicates were applied. The statistical analysis is not serious without it. However, in the caption/legend of “non-published” Table 3 and 4, it is written that “Three biological replicates were considered for each variety”. Each year too, I think. So, based on only three data, an example shows that the Tukey test derived that 2.327±1.021 is significantly different from 3.280±2.093. Sorry, but I cannot believe.
Overall, I do not think this manuscript is sound enough for the publication in Plants.
Reviewer 2 Report
The manuscript titled “Anthocyanin content and Fusarium mycotoxins in pigmented wheat (Triticum aestivum L. spp. aestivum): an open field evaluation” reports findings of some importance.
Overall, the manuscript is well written, however, there are some issues which are mentioned below.
In the title, the first letter of “Aestivum” should be small.
The abstract is somewhat well written provided with important data.
Keywords correspond to the aim.
The introduction is specific and focused on. The last paragraph of the “Introduction” described what the authors intend to do but this should be revised to make the objectives clear, robust and concise.
Materials and methods section is well written. However, some grammatical errors were spotted.
Results are quite interesting and analysis is strong; well written and explained.
Discussion confirmed results very well and is a logical explanation thereof.
Conclusion needs revision. The authors should give some recommendation on the basis of their findings.
Numerous stylistic errors were also spotted.
References are adequate and need to be crosschecked.
The language is up to the mark; however, some grammatical errors were spotted. In some cases, the authors used present tense to describe the results.
Reviewer 3 Report
Although I do not feel qualified to judge about the English language, I have noticed some typing / spelling errors.
For example:
line 3 (title) - word Aestivum shoul not be written in capital letter
line 36 - .... wheat resistance to abiotic and abiotic stress
line 148 - The plots measured 7 x 1,5 m2
general in the paper - in few places the words "cultivar", "variety" and "breeding line" are used as synonyms
e.g. in abstract (line 14) - Twelve Triticum aestivum L. spp. aestivum cultivars and in fact there is 9 cultivars and 3 breeding lines
Reviewer 4 Report
The authors presented an interesting result on an open field study on possible relationship between anthocyanin content and mycotoxin occurrence in pigmented wheat genotypes. The authors should address the comments below.
Line 241-244, sentence is difficult to understand. Correct accordingly
Line 281, “… and this is also consistent with Table 2.” Do you mean Table 2? Correct accordingly.
Line 356, add a comma after “varieties”
Line 380-384, Long sentence and very difficult to understand. Correct sentence
Line 418-421, I advise the authors write the initials as the names appear in the list of authors
Round 2
Reviewer 1 Report
OK